# Engineering the formation of spin-defects from first principles

Cunzhi Zhang[1], Francois Gygi [2] & Giulia Galli [1,3,4] ✉

The full realization of spin qubits for quantum technologies relies on the ability to control and design the formation processes of spin defects in semiconductors and insulators. We present a computational protocol to investigate the synthesis of point-defects at the atomistic level, and we apply it to the study of a promising spin-qubit in silicon carbide, the divacancy (VV). Our strategy combines electronic structure calculations based on density functional theory and enhanced sampling techniques coupled with first principles molecular dynamics. We predict the optimal annealing temperatures for the formation of VVs at high temperature and show how to engineer the Fermi level of the material to optimize the defect's yield for several polytypes of silicon carbide. Our results are in excellent agreement with available experimental data and provide novel atomistic insights into point defect formation and annihilation processes as a function of temperature.

Spin defects in wide bandgap semiconductors are promising platforms for several quantum technologies, including quantum photonics, and quantum sensing and communication[1,2]. Among spin qubit hosts, in recent years silicon carbide (SiC) has emerged as an ideal material, due to mature growth, doping and fabrication techniques[3,4], with qubits realized with silicon vacancies ($V_{Si}$) and nitrogen-silicon vacancy pairs ($N_C V_{Si}$), divacancies ($V_C V_{Si}$), and carbon antisite vacancies (CAV). The $V_C V_{Si}$ in SiC (which we denote as VV) has attracted particular interest, due to its optical addressability[5], a near-infrared spin-photon interface[6], long coherence times[7] and high-fidelity readout via spin-to-charge conversion[8]. While numerous studies of defects in SiC have focused on their physical properties, much less is known about their formation processes, whose control is critical for the integration of semiconductors hosting spin qubits within electronic and optical devices[1,3,4,9–12].

Defects in SiC are usually generated via implantation, irradiation or pulse laser, and by subsequent thermal annealing at high temperature[1,4]. Several experimental methods have been used to monitor defect formation, including electron paramagnetic resonance (EPR), photoluminescence (PL), and deep-level transient spectroscopy[13–18]. Recent progress has been reported in achieving spatial localization of defects[11], as well as in controlling their charge

state[19], performance and yield[14,20]. In the case of the VV, one of the most studied defects in SiC, it is well established that n-doping conditions are beneficial to its formation[16,21–23], and a lower bound for the annealing temperature ($T_{Ann}$) required to generate VVs has been estimated experimentally[16,24–27]. However, different experiments have reported different temperatures[8,15,24,26–33], with an optimal $T_{Ann}$ often quoted around 1150 K[24,25,34,35]. The experimental determination of activation and optimal annealing temperatures remains a challenging task, because these quantities are usually inferred from the intensity of EPR/PL signals which are affected by several factors, including the charge state and concentration of defects[36], Fermi-level position ($E_F$)[15,27], and specific synthesis conditions[37,38]. Recently, the pairing of $V_C$ and $V_{Si}$ into neutral VVs has been investigated theoretically, providing the first atomistic insight into the formation process[39].

However, as is the case for most point defects in semiconductors, our understanding of the VV formation mechanism at the atomistic level remains preliminary and qualitative. In particular, a relation between the host $E_F$ and $T_{Ann}$ has not yet been established, which is of great importance to control defects' formation, and an upper bound to $T_{Ann}$ is yet unknown. Moreover, the dynamics of VV[23,39], the conditions for the defect immobilization in the lattice and the effect of temperature on formation processes are only partially understood.

[1]Pritzker School of Molecular Engineering, University of Chicago, Chicago, IL, USA. [2]Department of Computer Science, University of California Davis, Davis, CA, USA. [3]Department of Chemistry, University of Chicago, Chicago, IL, USA. [4]Materials Science Division and Center for Molecular Engineering, Argonne National Laboratory, Lemont, IL, USA. ✉e-mail: gagalli@uchicago.edu

Addressing these open problems is difficult from an experimental standpoint, especially in the presence of limited microscopic resolution, and atomistic simulations are key tools to gain detailed insights.

Here we present a general computational protocol, based on first principles calculations, to study the formation of point defects in covalently bonded materials; in addition we provide specific predictions on optimal conditions for the formation of double vacancies in SiC. We focus on the cubic phase (3C-SiC) for its simplicity with only one type of lattice site, and we discuss implications of our results for hexagonal polytypes. We determine the preferred pathways leading to the VV formation and optimal values of $T_{Ann}$ and $E_F$, and we elucidate the interdependence of these parameters. Our results point at the importance of considering multiple charge states of defects, as well as of configurations that are not thermodynamically stable, for accurate predictions of formation pathways. On the other hand, the sampling of paths with different spin states has a negligible impact on our predictions.

## Results

### Computational strategy

We studied defect dynamics and transformations during the thermal annealing process following defect generation by e.g., particle irradiation. We considered several possible processes relevant to the formation of VVs in 3C-SiC, based on previous studies[21–23,39], and on our chemical intuition; they are summarized in Fig. 1. In addition, we considered dissociation processes, specifically CAV → $C_{Si}$ + $V_C$ (where $C_{Si}$ is an isolated carbon antisite) and VV → $V_C$ + $V_{Si}$, which involve multistep migrations of mono-vacancies (MV). We also considered the migration of CAV, a three-step process where $V_{Si}$ is an intermediate state: CAV is first converted to $V_{Si}$ (CAV → $V_{Si}$), followed by $V_{Si}$ migration; finally $V_{Si}$ is converted back to CAV ($V_{Si}$ → CAV). These dissociation and CAV migration pathways are not explicitly shown in Fig. 1.

We did not consider interstitial (e.g., Si or C interstitial), and substitutional (e.g., N substitution ($N_C$) or $C_{Si}$) defects; the former are expected to be annealed out once the paths described in Fig. 1 occur[22,23], and the latter are immobile at ~1000 K[40].

The simulation protocol used in our work is presented in Fig. 2. We studied the processes displayed in Fig. 1 using density functional theory (DFT) calculations with both the Perdew-Burke-Ernzerhof (PBE) and dielectric-dependent hybrid (DDH) functionals, and we considered several charge ($q$) and spin ($s$) states (see Methods). Specifically, we considered different $s$ states to determine the minimum energy path and energy barrier $E_b$ as a function of $q$, and for a given pathway, we obtained an effective barrier, $E_{b,EFF}$, as a function of the Fermi level $E_F$:

$$E_{b,EFF}(E_F) = \min_q \left\{ \Delta E_f(q, E_F) + E_b(q) \right\} \quad (1)$$

where $\Delta E_f(q, E_F) = E_f(q, E_F) - \min_q \{ E_f(q, E_F) \}$ is the formation energy difference relative to the most stable charge state, for a specific value of $E_F$, which in Equation (1) is treated as a parameter; $E_f$ is the formation energy of a defect in the initial state and $E_b$ denotes barriers between the initial and transition states. Note that $E_{b,EFF}$ is a continuous function of $E_F$, while $E_b$ exhibits steps at charge transition levels (see Supplementary Fig. 6). The expression of $E_{b,EFF}$ in Equation (1) assumes that charge state equilibration processes are faster than the transformation of defects into different configurations. We verified the validity of this assumption at high $T$ (~1000 K; see Supplementary Note 4). We emphasize that thermodynamically unstable $q$ states may participate and play an important role in defect transformation processes, since exploring those states may lead to lower effective barriers.

As mentioned above, the Fermi level is a parameter in Equation (1), and we estimated the experimental conditions that may lead to specific, desired values of $E_F$ based on charge neutrality conditions and the electronic properties of the system (see Supplementary Note 6).

We estimated the entropy change $\Delta S$ from the initial to the transition state by computing the difference in free energy barriers $\Delta G_b$ between 0 and 1500 K, where $G$ at 0 K is:

$$G(\xi^0, 0K) = \min_x U(x)|_{\xi(x) = \xi^0} \quad (2)$$

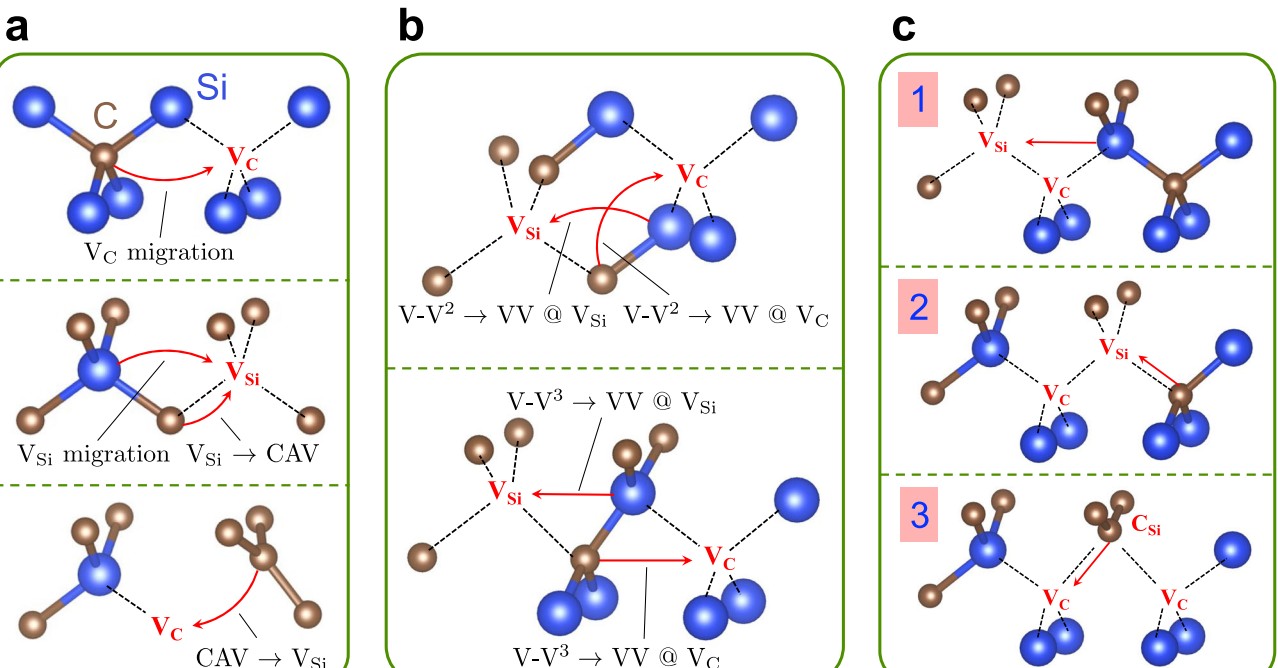

Fig. 1 | Investigated atomic pathways in 3C-SiC. a Monovacancy dynamics, including carbon ($V_C$) and silicon ($V_{Si}$) vacancy migration, and $V_{Si}$ and carbon antisite vacancy complex (CAV) inter-conversion. b Pairing of second (V-V²) and third (V-V³) neighbors $V_C$ and $V_{Si}$ vacancies to form a double vacancy VV. Only V-V up to third neighbors were considered, due to the size of our supercells. c VV migration path with the lowest barrier, where steps 1 to 3 are illustrated. $V_C C_{Si} V_C$ complex in step 3 is denoted as VCV.

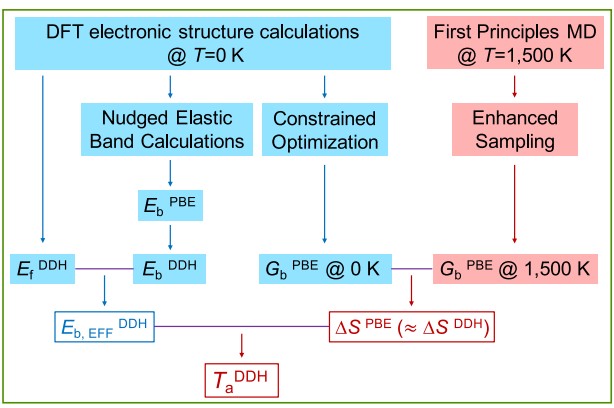

**Fig. 2 | Computational Protocol.** The calculations highlighted in blue (red) were carried out at zero (finite) temperature ($T$). We specify the functionals used in the calculations (PBE and DDH, see text) and the computed quantities: defect formation energies ($E_f$), energy and effective energy barriers ($E_b$ and $E_{b,EFF}$), Gibbs free energy barriers ($G_b$), entropy differences ($\Delta S$) and activation temperatures ($T_a$). We obtained $E_b^{PBE}$ using the nudged elastic band method, then corrected our results using the DDH functional to obtain $E_b^{DDH}$ (see Methods). Assuming error cancellation, we considered $\Delta S^{PBE} \approx \Delta S^{DDH}$ (see Supplementary Note 2).

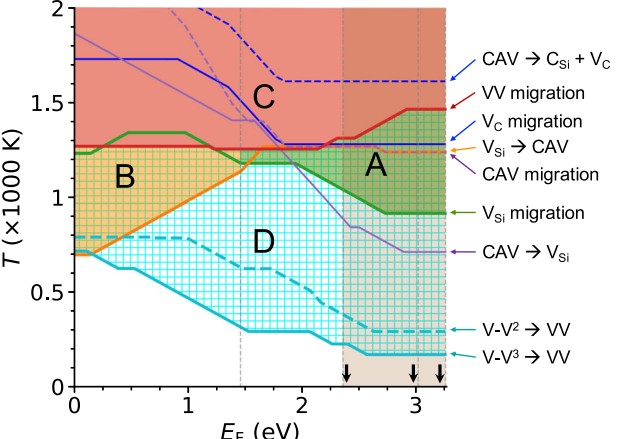

**Fig. 3 | Computed activation temperature ($T_a$) as a function of the Fermi level ($E_F$).** The Fermi level is referred to the top of the valence band. We indicate the processes on the right hand side of the figure via colored arrows; they are summarized in Fig. 1 and in the text (Computational strategy); the notation V-V$^{2/3}$ → VV refers to V-V$^{2/3}$ → VV @ V$_{Si}$ (Fig. 1b), a path with lower barrier than V-V$^{2/3}$ → VV @ V$_C$. The lines indicate the temperature, as a function of $E_F$, above which the process indicated on the right hand side is activated. Regions A, B, C and D where specific processes occur are described in the text. The vertical dashed lines indicate the $E_F$ at 1.46 eV and the conduction band minimum (CBM) of SiC polytypes. We computed the CBM of the various polytypes by aligning their respective valence band maximum; they are highlighted by black arrows; they are shown in increasing order of energy for 3C, 6H, and 4H of SiC. We extrapolated our results for $E_F$ higher than the CBM of 3C-SiC (shaded region), providing qualitative predictions.

and $\xi$ is a collective variable (the choice of collective variables is described in Methods and Supplementary Note 2); $U$ is the potential energy and **x** are atomic coordinates. We calculated $G$ at high temperature, specifically $T = 1500$ K, using first-principles molecular dynamics (MD) and the adaptive biasing force method (see Methods), and we estimated $\Delta S$ as $\Delta G_b/T$ (see Supplementary Note 2). Due to the computational cost, we obtained $\Delta S$ for only three paths (see Supplementary Note 2).

Once we obtained $E_{b,EFF}$ and $\Delta S$, we could compute the temperature $T_a$, above which a given process is thermally activated, and for which we used the harmonic transition state theory:

$$T_a = \left[ k_B \ln \left( \Gamma_0 \exp \left( \frac{\Delta S}{k_B} \right) / \Gamma \right) \right]^{-1} \times E_{b,EFF} \qquad (3)$$

where $\Gamma_0$ denotes an attempt frequency and $\Gamma$ a jump frequency. The values chosen for $\Gamma_0$, $\Gamma$ and $\Delta S$ are given in Supplementary Note 5. A simple sensitivity analysis, also in Supplementary Note 5, shows that in Equation (3) the prefactor is relatively insensitive to the choice of these values. In addition, we systematically investigated the effect of thermal expansion and that of entropy on computed activation temperatures, amounting to variations in $T_a$ of less than 10% (see Supplementary Note 5).

## Theoretical predictions

We start by presenting our results for 3C-SiC and we report our predictions for the activation temperature $T_a$ for various processes.

In Fig. 3 we show $T_a$ as a function of $E_F$, where lines indicate the values above which a given process can occur. We find that for all values of the Fermi level, the onset of $V_{Si}$ migration occurs at temperatures lower than those activating $V_C$ diffusion, consistent with the results of previous studies[16,21–23,25,35,41]. Above 1000 K, $V_{Si}$ can diffuse, and hence when migrating it may lead to the formation of VV. Interestingly, our calculations show that the pairing of mono-vacancies is facilitated by the Coulomb interaction between $V_{Si}^-$ and $V_C^{+1/+2}$. Indeed, we find that for $1.46 < E_F < 1.85$ eV, $V_{Si}^-$ and $V_C^{+1/+2}$ are respectively the most stable charge states of the two mono-vacancies (whether $V_C$ is in charge state +1 or +2 depends again on the Fermi level). It is important to note that for $E_F < 1.85$ eV, a simple consideration based on energy barriers $E_b$ would yield $T_a \sim 1500$ (1700) K as the temperature

required for a carbon vacancy to migrate in a stable charge state $q = +1$ (+2). However, upon computing effective barriers, we find a process with lower $T_a$ (as low as ~1300 K); such process involves intermediate charge states that are not thermodynamically stable but nevertheless allows for paths with lower barriers. Specifically, we find that thermal vibrations and changes in carrier density at high $T$ can cause a transition from $V_C^{+1/+2}$ to $V_C^{0/+1}$ charge states, and that the latter migrate through a path with a barrier lower than that of $V_C^{+1/+2}$, before returning to the original charge state (see Supplementary Notes 3 and 4).

The migration of $V_{Si}$ discussed above is a necessary but not a sufficient condition for the formation of VV. We find, in agreement with previous studies[16,21–23], that it is important to realize, at the same time, n-type conditions. In particular $E_F$ should be above 1.46 eV (see Fig. 3). Indeed, under p-type conditions ($E_F < 1.46$ eV), the $V_{Si}$ → CAV process is energetically favored over monovacancies diffusion (region B in Fig. 3) and $V_{Si}$ is trapped into the CAV complex and becomes immobile[21,22]. Once formed, CAV remains stable as the back conversion to $V_{Si}$ (CAV → $V_{Si}$), the CAV dissociation (CAV → $C_{Si}$ + $V_C$) and the CAV migration processes are all unlikely below 1500 K due to high free energy barriers. Instead, under n-type conditions ($E_F > 1.46$ eV) the migration of $V_{Si}$ is an energetically favored process and VV creation may occur (region A in Fig. 3). We note that in general, the higher $E_F$, the more favorable the conditions for VV formation for several reasons. Increasing $E_F$ leads to a lower MV migration barrier and increased mobility for $V_{Si}$, and to a higher (reduced) barrier for $V_{Si}$ → CAV (CAV → $V_{Si}$), leading to a lower probability of CAV formation and a higher probability of CAV conversion to $V_{Si}$. We also note that the barrier for the CAV → $V_{Si}$ process is always lower than that of the CAV → $C_{Si}$ + $V_C$ and of CAV migration processes. Hence, the CAV complex is stable under p-type conditions; instead, under n-type conditions, CAV can be annealed out once it is converted to $V_{Si}$, acting, in practice, as a reservoir of $V_{Si}$.

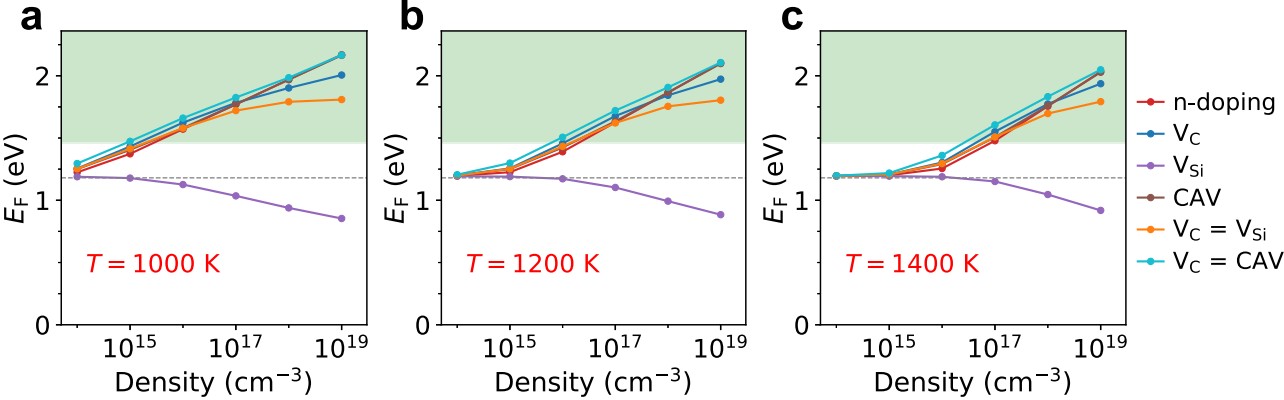

**Fig. 4 | Fermi level ($E_F$) as a function of defects density in 3C-SiC.** The Fermi level is referred to the top of the valence band. We show results for different temperatures (1000 K **a**, 1200 K **b**, and 1400 K **c**); we consider, separately, initial n-doping of the sample, and specific concentrations of carbon ($V_C$) and silicon ($V_{Si}$) vacancies, and carbon antisite vacancy complexes (CAV); we also consider two additional cases: (i) same concentration of $V_C$ and $V_{Si}$ ($V_C = V_{Si}$); (ii) same concentration of $V_C$ and CAV ($V_C = $ CAV). The dashed line indicates the value of mid-gap for 3C-SiC; the green-region for $E_F > 1.46$ eV indicates favorable conditions for the formation of the double vacancy in the range of temperatures between 1000 and 1300 K (see text).

An additional condition for the formation of VVs is an annealing $T$ below 1300 K. We find that VV can migrate for $T > \sim 1300$ K and will likely either form large complexes (e.g., VV + $V_C$)[42] or diffuse and eventually move to the surface of the sample (region C in Fig. 3). These processes undermine the stability and abundance of double vacancies. In addition we find that $V_C$ is immobile below 1300 K, which is overall a favorable condition for VV formation. Indeed we expect $V_C$ to be abundant in experimental samples, due e.g., to a formation energy lower than that of $V_{Si}$ and other point defects, and it can be incorporated in a VV + $V_C$ cluster if it migrates. We also find that VV dissociation (VV → $V_C + V_{Si}$) is unlikely to occur below 1300 K. Therefore, we suggest that $T_{Ann}$ should be < 1300 K for optimal yield, stability and localization of VV defects.

Note that larger vacancy clusters can be formed by incorporating mobile $V_{Si}$ into VV[21]. Note also that the diffusion of $C_{Si}$, and possibly of other dopants, can be activated by the migration of $V_{Si}$[43], and could lead to the formation of additional complexes, by reacting with VVs. The overall effective barrier of such processes is expected to be higher than, and in some cases possibly comparable to, that of $V_{Si}$ migration. The undesirable formation of these complexes may be mitigated by reducing the concentration of $V_{Si}$. Therefore, adopting a high concentration of $V_C$ and a low amount of $V_{Si}$ can be a useful strategy not only to facilitate the creation of VV but also to stabilize them. Unfortunately, charge-state engineering[12,14,20] is not an effective tool to hinder the aggregation of VV with other defects because the most stable state of VV is neutral for $E_F$ above mid-gap.

Other defects of potential concern for the stability and formation of the VV are single interstitials ($C_i$ and $Si_i$). For example, $C_i$ could be re-emitted from $C_i$ clusters at high $T$, and subsequently aggregate with VV. Fortunately, $C_i$ emission is unlikely to occur below 1300 K, according to previous DFT calculations (barrier $> \sim 4$ eV)[42,44,45]. Nonetheless, VV could be annihilated by the presence of $C_i$ if some weakly bonded $C_i$ clusters turn out to be present in the sample.

To estimate the optimal annealing $T$ in the range of (1000 and 1300) K, we consider the dependence of the Fermi level on doping densities. In 3C-SiC, maintaining $E_F > 2$ eV requires a rather large doping density $> \sim 10^{18}$ cm$^{-3}$ (see Fig. 4). Therefore, it is conceivable that a desirable Fermi level range is $1.46 < E_F < 2$ eV, over which $T_a$ for $V_{Si}$ migration is a constant, roughly equal to 1200 K. For $T_{Ann}$ between 1200 and 1300 K the Fermi level would be lower, for a fixed doping density, than in the range 1000–1200 K, hence possibly leading to $V_{Si}$ trapping at CAV defects. Therefore we conclude that the optimal $T_{Ann} \sim 1200$ K.

So far, we have identified a suitable range of $T_{Ann}$ under n-type conditions, (1000, 1300) K, with an optimal value of 1200 K. However, as shown in Fig. 3 (region D), there exist conditions at which VV may form below 1000 K, as long as there are V-V defects present in the sample. Our results indicate that in general, as $V_{Si}$ and $V_C$ approach each other, the barrier of $V_{Si}$ migration towards $V_C$ would decrease, facilitating the pairing of V-V to form VV. These results may help understand the conditions required for VV formation in small SiC nanoparticles (with diameter less than 10 nm), observed at lower $T_{Ann}$, e.g., $\sim 440$ K[46], than in the bulk, since in nanoparticles MV separation distances are usually smaller. However, it is worth mentioning that the recombination of close interstitial-vacancy complexes and the interstitial migration process could involve low barriers, $< \sim 1.5$ eV in SiC, according to ref. 44,47; using these barriers the activation temperatures for these processes are estimated to be $< \sim 500$ K, i.e., within a range of $T$ where the pairing of V-V to form VV may also occur. Hence, we expect the recombination of V-V with interstitials and its pairing to be competing processes below 500 K, and interstitials may have the adverse effect of reducing the VV formation from V-V pairing.

We now turn to exploring how conditions identified above for VV formation may be achieved experimentally, by controlling for example the Fermi level and density of defects. In addition to the electronic properties of the system, $E_F$ depends on $T$, initial sample doping and of course defect density (see Supplementary Note 6) which, at each given time of the annealing process is the most elusive parameter. The spatial distribution of defect density may be non-uniform and it depends on the specific dose and energy of particles used during the bombardment of the sample[1,4,12,38]. In spite of these uncertainties, it is interesting to obtain a qualitative estimate of the doping conditions necessary to achieve the desirable Fermi level values for the formation of VVs. In Fig. 4, $E_F$ is calculated at several $T$ and for various doping and defect densities. We find that the presence of $V_C$ and/or CAV would induce n-doping while the presence of $V_{Si}$ would induce p-doping in the sample. Hence, the required condition to reach $E_F > 1.46$ eV within (1000, 1300) K, is that at least one of the following concentrations– n-doping (e.g., [$N_C$]), C vacancies [$V_C$] or antisite [CAV]$^-$ be larger than $10^{16}$ cm$^{-3}$.

Note that p-doping conditions induced by the presence of $V_{Si}$, which are unfavorable for the formation of VVs, may be compensated by the presence of $V_C$ or CAV of comparable amounts (see Fig. 4). Further, the $V_{Si}$ to CAV conversion process ($V_{Si}$ → CAV), although it renders $V_{Si}$ less mobile, helps to increase $E_F$ which in turns facilitates VV formation. These results emphasize the complex, interdependent role

of multiple defects in tuning $E_F$ and ultimately leading to the formation of VVs.

We note that the concentration of vacancies varies with annealing time and vacancies may be easily annealed out at high $T$. In addition, large amounts of residual vacancies may degrade the material's quality and hence its performance for quantum applications. Therefore, engineering $E_F$ by means of a large vacancy concentration may not be a good strategy. Nonetheless, at the beginning of the annealing treatment, vacancies can be present even at temperatures higher than their respective activation temperature for migration or conversion. Under these circumstances, the presence of vacancies would definitely affect the position of $E_F$. To characterize and further understand the impact of such transient species during the annealing process, we include the results for relevant vacancies at high temperatures in Fig. 4.

We observe that the proposed n-type conditions for efficient VV formation at high $T$, is also favorable to stabilize the desired $VV^0$ at low $T$; the presence of, e.g., $N_C$ or $V_C$ could increase the $E_F$ (see Supplementary Fig. 4) to a region where the $VV^0$ is stable[48,49]. However, we note that it is conceivable that there may be spin-defects or hosts, where the required doping conditions for the optimal qubit formation at high $T$ and its charge-state stabilization at low $T$ are different. In those cases, after qubit formation, further engineering of $E_F$ may be needed via, e.g., gating or additional doping.

Further, we discuss the VV formation properties in hexagonal (hex) polytypes, e.g., 4H-SiC. The extension of our results for 3C-SiC (where only $k$-sites are present) to hexagonal lattices (where both $h$- and $k$-sites are present) should be considered as a qualitative prediction (shaded region in Fig. 3). In hexagonal samples, the variation in stability and barriers of defects occupying different lattice sites is negligible, compared to the energy scale of ~ several eV of most barriers computed in our calculations[16,18,21,23,41]. The position of the valence band maximum (VBM) is nearly the same in cubic and hexagonal SiC, while that of the conduction band minimum (CBM) is higher in hex-SiC[50] (see Fig. 3). We find that the creation of VVs is more facile in hex-SiC; indeed the conditions of regions A (corresponding to formation and stability of VV) and D (corresponding to the pairing of nearby vacancies) can be obtained in a slightly wider range of temperatures than in 3C-SiC (for values of the Fermi level attainable in hex-SiC) and, importantly, for lower doping densities. For example, in 4H-SiC, under intrinsic conditions, $E_F$ ~ 1.6 eV is larger than 1.46 eV, and with a moderate n-doping > $10^{15}$ cm$^{-3}$, it may be increased above 2.0 eV at 1200 K (see Supplementary Fig. 5). We note that to synthesize VV in hex-SiC, it is beneficial to use samples at near intrinsic conditions, where the $VV^0$ is stable[48,49]. We predict that an appropriate $T_{Ann}$ for hex-SiC is in the range of (900, 1300) K, with an optimal value around 1200 K. We note that the charge-state equilibration process is slower in hex- than 3C-SiC (see Supplementary Fig. 3). Therefore, in Supplementary Fig. 6, we show activation temperatures determined from the most stable charge-state at a given $E_F$. By comparing Fig. 3 and Supplementary Fig. 6, we find that our qualitative predictions of the VV formation properties in hex-SiC are the same, whether using effective barriers or the data of Supplementary Fig. 6.

Our predictions are in excellent agreement with several experimental observations. To synthesize VV, most experiments adopted $T_{Ann}$ in a range of (1050, 1350) K, consistent with our prediction of annealing $T$ of (1000, 1300) K in 3C-SiC and (900, 1300) K in hex-SiC. Experimentally, the optimal $T_{Ann}$ was determined by PL or EPR maximum intensities and found to be ~ 1150 K, in agreement with our calculations of ~ 1200 K. We emphasize that depending on the experimental setup, the decrease in signal above ~ 1150 K is not necessarily related only to changes in VV concentration. We predict VV can be stable up to 1300 K, above which its density decreases due to diffusion. This is consistent with the significant drop of VV signals in experiments as $T$ > 1300 K[15], and with the highest PL and EPR intensities detected at 1300 K[16,17].

In closing, we note that the computational strategy adopted here to investigate the formation of VV and the results obtained in our work are useful to inform the investigation of other spin-defects in semiconductor hosts. For example, the formation of the $N_CV_{Si}$ (denoted as NV) center in SiC is also expected to be facilitated by the migration of $V_{Si}$ but in nitrogen-doped samples[51,52]. Therefore, we expect that the favorable $E_F$-$T$ conditions identified here for the creation of VV should be partially applicable to the NV center as well. Indeed, an annealing $T$ ~ 1200 K was adopted experimentally for NV formation[51,52], close to the predicted optimal value for the VV. Moreover, to guide and optimize the NV synthesis, our results can be used to estimate the desired $E_F$ to stabilize $V_C^0$ instead of $V_C^{+1/+2}$, and hence mitigate the aggregation of NV (negatively charged for n-type conditions[48]) and $V_C$. In addition, our computational strategy can be used to compute the NV migration barrier to design the proper annealing temperature; our results suggest the need to elucidate the effects of nitrogen doping on the system $E_F$ and on the induced reactions, e.g., the NV and $N_C$ aggregation. Importantly, our results indicate that NV should be easier to form in hex-SiC than 3C-SiC, as the VV, due to the higher $E_F$.

## Discussion

By combining DFT calculations with semilocal and hybrid functionals, nudged elastic band, and first principles MD simulations, we obtained a detailed, atomistic description of the VV formation process in 3C-SiC. We computed energy barriers and activation temperatures for multiple defects and pathways as a function of the Fermi level $E_F$. We then identified favorable conditions for the formation of VVs and discussed how suitable values of $E_F$ can be obtained via careful tuning of doping or defects densities. Our calculations show that one should use n-doped samples with $E_F$ > 1.46 eV during annealing, to ensure the stability of single $V_{Si}$, and $T_{Ann}$ > ~ 1000 K to activate $V_{Si}$ migration for aggregation with $V_C$. Further, $T_{Ann}$ should be lower than ~ 1300 K to suppress VV diffusion, thus ensuring its stability and immobilization, with the optimal $T_{Ann}$ estimated to be ~ 1200 K. However, VV can also be created at lower $T$ from neighboring $V_C$-$V_{Si}$ pairs; these may be present after irradiation or implantation, and may be prominent in SiC nanostructures, suggesting that the formation of VVs in small nanoparticles should occur at lower $T$ than in the bulk. Our findings also suggest that VV signals may be detected at low annealing temperatures, which however should not be interpreted as lower bounds for $V_{Si}$ diffusion. Moreover, we predict that VV formation in hex-SiC can be more facile than in 3C, due to a larger band gap and higher CBM position, which allow for the use of lower doping densities and lead to a slightly broader range of favorable annealing $T$. Our results are in excellent agreement with experiments, while providing new and improved understanding of formation mechanisms at the atomistic level. The knowledge obtained here may benefit the controlled fabrication and device integration of VV, assisting its applications for quantum technologies.

Importantly, the computational protocol and strategies developed here, based on first principles calculations, are general and can be readily extended to investigate defects in other covalently bonded materials. Multiple paths with different charge states should be considered to understand point defect formation processes, taking into account thermodynamically unstable ones, which may facilitate the exploration of low barrier paths at high $T$. Our findings show that it is key to conduct calculations of effective barriers as a function of the Fermi level, which itself depends on $T$, and not only of barriers between thermodynamically stable states. In addition, it is critical to consider not only formation but also annihilation pathways to obtain faithful predictions of formation processes. Unexpectedly, we found that although important for accurate quantitative predictions, thermal expansion and entropic contributions are not critical to determine general trends of activation temperatures for different paths.

One important problem that remains to be addressed is the influence, on defects' formation, of the specific synthesis procedures, e.g., by irradiation of the sample. Using our computed energy barriers as input, one can simulate real-time defect evolution, e.g., via kinetic Monte Carlo methods, which could then provide information about optimal annealing times. These possible directions are worthy of future explorations.

## Methods

### Density functional theory calculations

We performed DFT calculations using the Qbox[53] and the Quantum Espresso[54] codes. We used the PBE[55] and DDH[56] (15% exact exchange) functionals, optimized norm-conserving Vanderbilt pseudopotentials[57], a plane-wave kinetic energy cutoff of 60 Ry. We conducted calculations in 216 atom supercells with lattice constant 4.416 Å, and with either the $\Gamma$ point or a $2 \times 2 \times 2$ Monkhorst-Pack (MP) grid to sample the Brillouin zone. The lattice constant was determined by first-principles MD (FPMD) simulations in the NPT ensemble at 1500 K at the PBE level of theory. We considered structural relaxations as converged when residual forces on atoms were < 0.01 eV/Å. We considered charge state $q$ from -2 to 2 for all defects, expect for $V_C$ where $q$ = 0, 1 or 2; spin state $s$ = [S, T] ([D, Q]) for even (odd) number of electrons, where S: singlet, D: doublet, T: triplet, Q: quartet. We chose not to employ empirical force-fields, which would have allowed for the use of larger supercells, as they are not appropriate to simulate $q$ and $s$ degrees of freedom; in addition we found that in several cases many of the popular force-fields used for SiC cannot reproduce DFT results.

### Nudged elastic band calculations

We carried out climb image nudged elastic band (CI-NEB) simulations at the PBE level with a $2 \times 2 \times 2$ MP grid, by coupling Qbox with the PASTA[58] code. We used spring constants of 2 eV/Å$^2$ and force tolerance of 0.02 eV/Å. We determined the most stable spin state among [S, T] or [D, Q] for each NEB image at a given charge state $q$; the corresponding total energies and atomic forces were then used to update NEB images to determine the minimum energy path and energy barriers $E_b$. In this way, $E_b$ is only a function of $q$. For most pathways studied here, the most stable spin state remain the same along the whole path; for those paths for which we observed a change of spin states, we found that the energy splitting between different spin states at the transition-state is generally small, i.e., less than 10% of $E_b$.

We then computed total energies for converged images, at the PBE and DDH level of theory using only the $\Gamma$ point[18]. We denote the barriers obtained in this way as $E_b^{PBE}$ @ $\Gamma$ and $E_b^{DDH}$ @ $\Gamma$. We computed the correction to apply to PBE results in order to estimate DDH barriers as: [$E_b^{DDH}$ @ $\Gamma$ − $E_b^{PBE}$ @ $\Gamma$]. We added such correction to $E_b^{PBE}$ @ 222 (barriers computed with the $2 \times 2 \times 2$ MP grid) to obtain $E_b^{DDH}$ @ 222. Here, we assumed that the minimum energy paths at the PBE and DDH level of theory are similar; energy difference calculated with the $\Gamma$ point differed only slightly from those obtained with the $2 \times 2 \times 2$ MP grid. The results reported in the main text were obtained with $E_b^{DDH}$ @ 222.

### Formation energy calculations

The formation energy of defect X in charge state $q$, $E_f(X^q)$, was computed as:

$$E_f(X^q) = E_{tot}(X^q) - E_{tot}(SiC) - n_C \mu_C - n_{Si} \mu_{Si} + q E_F + E_{corr}(X^q) \quad (4)$$

where $E_{tot}(X^q)$ is the total energy of a SiC supercell with $X^q$; $E_{tot}(SiC)$ is the total energy of the pristine SiC supercell; $\mu_C$ and $\mu_{Si}$ are chemical potential of C and Si; $n_C$ and $n_{Si}$ are number of added (+) or removed (−) C or Si atoms to form X, respectively; $E_F$ is the Fermi energy referred to the VBM; $E_{corr}(X^q)$ is the energy correction for spurious electrostatic interactions present in supercell calculations.

Using relaxed configurations at the PBE level of theory, we computed the total energy and electrostatic potential using Quantum Espresso and the DDH functional. We used a $2 \times 2 \times 2$ MP grid. We obtained $E_{corr}$ using the method developed by Freysoldt, Neugebauer, and Van de Walle[59]. We used a dielectric constant equal to 9.72. The chemical potential $\mu_C$ was calculated as the energy per atom in diamond; $\mu_{Si}$ was calculated as $\mu_{SiC} − \mu_C$, where $\mu_{SiC}$ is the energy per formula unit in bulk SiC. The results are shown in Supplementary Fig. 1.

Finally, binding energies between defects are required to compute the barriers of the CAV/VV dissociation processes. We estimated the $C_{Si}$ and $V_C$ binding energies as ~1 eV from previous studies[21,22]. We directly computed the $V_C$ and $V_{Si}$ binding energies, which are ~3 eV for $E_F$ near the mid-gap of 3C-SiC.

### Enhanced sampling calculations

We computed free energies of defect transformations by coupling the Qbox and SSAGES[60] codes. We used Qbox to perform FPMD in the NVT ensemble and the adaptive biasing force method[61] in SSAGES to calculate free energy gradients. We utilized the collective variable (CV) $\xi$[22]:

$$\xi = \left( \mathbf{R} - \frac{1}{M} \sum_{i \in \text{gate atoms}} m_i \cdot \mathbf{R}_i \right) \cdot \mathbf{e}_{\text{projection}} \quad (5)$$

where $\mathbf{R}$ is the coordinates of the moving atom; $m_i$ is the mass of the $i^{th}$ gate atom; $\mathbf{R}_i$ is the coordinates of the $i^{th}$ gate atom; $M$ is the total mass of the gate atoms; $\mathbf{e}_{\text{projection}}$ is the unit projection vector (see Supplementary Fig. 2).

We carried out free energy calculations for the three processes presented in Supplementary Fig. 2, where the definition of CVs and gate atoms is specified. For each path, we performed FPMD simulation at 1500 K using a time step of 1 fs, for ~370 ps. For computational efficiency, we used the PBE functional, 40 Ry kinetic energy cutoff and the $\Gamma$ point; we considered defects only at $q$ = 0 and $s$ = T in our MD simulations.

To elucidate the effect of $T$ on barriers, we also computed free energy profiles at 0 K. We used the PBE functional, 40 Ry kinetic energy cutoff and the $\Gamma$ point for consistency. We used the Sequential Least SQuares Programming method[62] in the SciPy package to carry out constrained optimizations along one-dimensional CVs.

## Data availability

The data that support this study will be made available through Qresp (https://qresp.org/).

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

## Acknowledgements
We thank Yu Jin, Elizabeth M.Y. Lee, Joe Heremans and Marco Govoni for useful discussions. The design, implementation and execution of the computational strategy adopted in this work, as well as two of the main codes used in this work, Qbox and SSAGES, were supported by MICCoM, as part of the Computational Materials Sciences Program funded by the U.S. Department of Energy, Office of Science, Basic Energy Sciences, Materials Sciences, and Engineering Division through Argonne National Laboratory. The extensive analysis of experimental data that led to the choice of materials and experimental conditions was supported by QNEXT. This research used computational resources at the University of Chicago's Research Computing Center and the Argonne Leadership Computing Facility located at Argonne National Laboratory. Computer time at Argonne Leadership Computing Facility was provided by the Department of Energy's ASCR Leadership Computing Challenge (ALCC).

## Author contributions
C.Z., F.G., and G.G. designed the calculations. C.Z. performed the calculations. All authors contributed to the data analysis and manuscript writing.

## Competing interests
The authors declare no competing interests.
