## [Peer Review File · Nature Communications]

REVIEWER COMMENTS

Reviewer #1 (Remarks to the Author):

This paper is a logical continuation of the work reported in *Natur. Commun.*, 12, 6325 (2021) by the same group. It presents a “computational protocol” to predict the optimal thermal and doping conditions for the formation of the divacancy defect in SiC, which is a prime candidate for realizing solid state qubits. The suggested framework is quite impressive and it is gratifying to see that theory has come to the level where useful advice can be given for the design of semiconductor technologies. The predictions born out of the paper agree well with what is known from experiment, while atomistic insight is provided for the reasons.

The suggested computational protocol contains high level methods and well-founded approximations, and represents a definitive improvement with respect to the preceding paper mentioned above, by abolishing the faulty force-field MD simulations.

I think, the real significance of the paper stems from the build-up of the methodology. The authors demonstrate that it works well for this particular problem, since they can reproduce experimental results well but, from the practical point of view, nothing dramatically new is provided. However, I consider the “theoretical” point made in the paper, about the importance of metastable state, very important. It reinforces a similar warning, published very recently in *Natur Commun.*

The paper might be more appropriate for *npj Comput Mater*, but I don't have a strong feeling about that.

The paper is very well written. Before publication I have one concern though. The authors neglect the possibility for the motion of C_{Si}- antisites, even though it is known that the motion of V_{Si} can promote that. (The review paper referenced by them claims the immobility without the presence of an excess of V_{Si}.) This point needs more discussion.

Reviewer #2 (Remarks to the Author):

The authors present exciting results capturing optimal annealing temperatures for the formation of divacancies in 3C-SiC and optimal tuning of the Fermi level for high-yield defect formation. The results are certainly noteworthy and are relevant to the field of spin qubits for quantum technologies based on spin defects in semiconductors and insulators. I would support publication of the work in Nature Communications after the following comments are addressed.

1. Why is the range ~ 0.01 Hz - 1 Hz from refs. 13 and 14 on p. 7 in the fourth paragraph of the supplementary material not rescaled by the ratio of the Debye frequencies of Ga₂O₃ (<https://aip.scitation.org/doi/pdf/10.1063/1.4916078>) and 3C-SiC (<https://journals.aps.org/prb/pdf/10.1103/PhysRevB.54.1791>), or some similar measure of vibrational frequencies? Instead it seems the geometric mean of ~ 0.01 Hz and 1 Hz was taken to obtain the value 0.1 Hz even though both references considered Ga₂O₃ and not 3C-SiC.

2. I'm not sure I understand the argument made in the third paragraph on p. 9. The argument appears to be saying that the barriers decrease as the distance between monovacancies increases and that as a result nanoparticles should have smaller annealing temperatures than bulk due to the smaller associated distances. Shouldn't it be that if nanoparticles exhibit smaller distances between monovacancies than in the bulk, the annealing temperatures should be higher if the barrier decreases with increasing distance?

3. The work appears to propose a general framework for investigating the synthesis of point-defects at the atomistic level, but the benchmarking focuses on a single defect the VV. Would it be possible to comment on whether or show that the formalism is indeed general (ie. by including more examples in 3C-SiC or other hosts, for example a comparison of the predicted annealing temperature for the formation of carbon vacancy-carbon antisite complexes in 3C-SiC and those found in the literature - <https://journals.aps.org/prb/abstract/10.1103/PhysRevB.101.184108>)

4. Further, regarding the generalizability of the results, would it be possible to comment on why the use of a PBE lattice constant is valid for obtaining barriers at the DDH level of theory. Essentially, the title appears a bit misleading if the first principles theory being used is not self-consistent.

5. Expanding on the previous comment, would it be possible to further elaborate on why we would expect error cancelations at the PBE and DDH levels of theory (from the supplementary material on p. 3 in the third paragraph)? (ie. Is it possible to show for some small supercell or for the unit cell that total energies calculated following perturbations to the atomic positions indeed exhibit the desired error calculations at the PBE and DDH levels of theory?)

Reviewer #3 (Remarks to the Author):

Silicon carbide (SiC) host prototypical color centers relevant for applications in quantum technology. The potential of SiC color centers not only hinges on the opportunities SiC provides as a mature electronic material, but also on the ability of their controlled intentional creation at the desired location to allow for single color center operation in place of ensembles.

The manuscript by Cunzhi Zhang Francois Gygi and Giulia Galli addresses the formation and annealing kinetics of prototypical color centers in Silicon carbide, namely the silicon vacancy, carbon antisite-vacancy complex, and the di-vacancy in SiC. They focus on the 3C polytype and investigate the activation barriers of prototypical mechanisms consisting of nearest neighbor and next neighbor hops in the frame work of density functional theory. The calculations involve the semilocal PBE-functional and the DDH hybrid functional and are conducted for using standard techniques for finding saddle points and energy barriers. For specific mechanisms they also include a finite temperature relevant for kinetic processes based on enhanced sampling within ab initio molecular dynamics.

The formation and annealing kinetics of intrinsic defects in SiC has been investigated for more than twenty years both by theory and experiment and has seen renewed interest with SiC color centers becoming candidates for qubits and single-photon sources in particular

as open questions regarding vacancy complexes including the di-vacancy exist. The present manuscript follows an earlier work of the authors published last year in Nature Communications on the same topic focusing more on explorative ab initio molecular dynamics simulations. The present manuscript in large part confirms the results of earlier works regarding barriers and the Fermi level dependence of annealing mechanisms for the vacancies and the carbon antisite-vacancy complex adding additional insight in case of the di-vacancy. As key result they summarize this knowledge in a diagram showing the activation temperature for specific formation or annealing mechanisms vs. the Fermi-level. Based on this diagram and a discussion of the temperature dependence of the Fermi level as a function of defect density they conclude that an n-type Fermi level together with a certain temperature provide optimal conditions for the formation of quantum bits in 3C SiC.

Given that the present, well-written manuscript largely builds on the previous work of the authors already published in Nature Communications and the earlier work by other groups, and the progress achieved I think the paper reflects the outcome of a regular article. In particular, important results of an in-depth analysis of the formation and annealing kinetics concerning a prototypical qubit center, the NV-centers in diamond were published as a regular article. In this light I do not support the publication of the present work in Nature Communications. Furthermore, the manuscript and conclusion contains a few issues that should be addressed:

1) Although n-type conditions favor the formation or stability of the desired color centers, the conditions to stabilise the relevant active charge may not be compatible with n-type conditions. In the diagram in Figure 3 should also contain the information on the stability range of the relevant active charge states of the silicon vacancy and the di-vacancy. Only in the range of the Fermi level where the active charge state is stable, the proposed Fermi-level engineering provides an advantage. With this in mind

2) In Fig. 3. the combination of nearby vacancies to di-vacancies with very low activation temperature is shown. Although these reactions certainly takes place at low activation temperature it requires the generation of such pairs as a result of previous implantation - vacancy migration can be excluded as a source at these temperatures. Under these conditions nearby interstitials are also available and the recombination of vacancies with interstitials is a competing process. For this reason Fig 3. should also include the activation temperature for the interstitial migration and vacancy-interstitial recombination.

The arguments in the discussion should be refined correspondingly

3) The authors argue that charge state equilibrium occurs on timescales that are shorter than the timescale between successive migration events. The authors' estimate in support of this argument (cf. Note 3 of the supplementary material) does not include the dependence on the Fermi-level position of the carrier density, which is exponential. So while this argument is certainly true, when the free

carrier density is high, vanishing free carrier densities in high purity semi-insulating SiC material does, indeed, lead to very slow recombination dynamics. In order to complement their argument the authors should plot the estimate for the recombination rate as a function of the Fermi-level position both for the 3C and 4H prototypes. In the expected case that there is a range of Fermi-levels

4) In Fig. 4 high concentrations of silicon vacancies are assumed as dominating defects at 1200K and 1400K. According to Fig. 3, however, the activation temperature for the conversion into the carbon antisite-vacancy complex is indicated to be below 1400K. This seemingly contradicts the conditions assumed in Fig 4. The main text does not provide in depth explanation in this respect. The authors should discuss the implications more rigorously in the text and resolve the seeming contradiction.

While directly after irradiation with high particle doses high vacancy concentrations are readily created, annealing at high temperature rapidly reduces these concentrations through interstitial-vacancy recombination or vacancy out-migration to thermal equilibrium. This can only be achieved by reducing the temperature below the activation temperature of vacancy migration. Is it reasonable to forcefully maintain a high vacancy concentration throughout the annealing to produce a higher di-vacancy concentrations and at the same time assume to obtain a high-grade material for quantum applications? The authors should answer this problem in the discussion of Fig. 4. I have doubts that the proposed route of Fermi-level engineering via high vacancy concentrations is a successful strategy.

Response to reviewers

We thank the reviewers for their thoughtful suggestions and comments. We have revised our manuscript by taking into account all of their concerns. In the following we present a point-by-point reply to all the comments of the reviewers.

In addition, we added a discussion of the CAV migration process in Fig. 3 for completeness. All changes relative to the original manuscript are shown in blue in the main text and in the SI.

Reviewer #1 (Remarks to the Author):

This paper is a logical continuation of the work reported in Natur. Commun, 12, 6325 (2021) by the same group. It presents a “calculational protocol” to predict the optimal thermal and doping conditions for the formation of the divacancy defect in SiC, which is a prime candidate for realizing solid state qubits. The suggested framework is quite impressive and it is gratifying to see that theory has come to the level where useful advice can be given for the design of semiconductor technologies. The predictions born out of the paper agree well with what is known from experiment, while atomistic insight is provided for the reasons.

We thank the reviewer for the positive comments.

The suggested calculational protocol contains high level methods and well-founded approximations, and represents a definitive improvement with respect to the preceding paper mentioned above, by abolishing the faulty force-field MD simulations.

We thank again the reviewer for the positive comments.

I think, the real significance of the paper stems from the build-up of the methodology. The authors demonstrate that it works well for this particular problem, since they can reproduce experimental results well but, from the practical point of view, nothing dramatically new is provided. However, I consider the “theoretical” point made in the paper, about the importance of metastable state, very important. It reinforces a similar warning, published very recently in Natur Commun.

We would like to emphasize that not only our computational protocol is novel and was never applied before to the study of defects’ formation in semiconductors, as recognized by the reviewer. In addition, the results of our simulations led to a wealth of quantitative predictions on how to engineer the Fermi level and design optimal conditions in experiments, which again were never reported before. Hence from a fundamental point of view, many new results are presented in our

paper on the physics of point defects in SiC, which hopefully will contribute to designing useful practical realization of synthetic routes.

The paper might be more appropriate for npj Comput Mater, but I don't have a strong feeling about that.

The paper is very well written. Before publication I have one concern though. The authors neglect the possibility for the motion of CSi^{-} antisites, even though it is known that the motion of VSi can promote that. (The review paper referenced by them claims the immobility without the presence of an excess of VSi .) This point needs more discussion.

We added a discussion of the motion of the C_{Si} antisite and its influences on the VV formation processes on page 9:

“Note also that the diffusion of C_{Si} , and possibly of other dopants, can be activated by the migration of V_{Si} [43], and could lead to the formation of additional complexes, by reacting with VVs. The overall effective barrier of such processes is expected to be higher than, and in some cases possibly comparable to, that of V_{Si} migration. The undesirable formation of these complexes may be mitigated by reducing the concentration of V_{Si} . Therefore, adopting a high concentration of V_C and a low amount of V_{Si} can be a useful strategy not only to facilitate the creation of VV but also to stabilize them.”

Reviewer #2 (Remarks to the Author):

The authors present exciting results capturing optimal annealing temperatures for the formation of divacancies in 3C-SiC and optimal tuning of the Fermi level for high-yield defect formation. The results are certainly noteworthy and are relevant to the field of spin qubits for quantum technologies based on spin defects in semiconductors and insulators. I would support publication of the work in Nature Communications after the following comments are addressed.

We thank the reviewer for the positive comments.

1. Why is the range ~ 0.01 Hz - 1 Hz from refs. 13 and 14 on p. 7 in the fourth paragraph of the supplementary material not rescaled by the ratio of the Debye frequencies of Ga_2O_3 (<https://aip.scitation.org/doi/pdf/10.1063/1.4916078>) and 3C-SiC (<https://journals.aps.org/prb/pdf/10.1103/PhysRevB.54.1791>), or some similar measure of vibrational frequencies? Instead it seems the geometric mean of ~ 0.01 Hz and 1 Hz was taken to obtain the value 0.1 Hz even though both references considered Ga_2O_3 and not 3C-SiC.

The range of (0.01, 1) Hz refers to the values of the jump-frequency (Γ) (see Note 5 in SI), above which a given process is activated. For instance, we consider that a given process is activated once

it can occur or is observed in more than 10 instances during a total observation time of 10 s, i.e. $\Gamma \geq 1$ Hz; hence Γ does not depend on the systems. Hence the threshold range of Γ , i.e. (0.01, 1) Hz, is a system independent choice and no rescaling is necessary. In our study, we chose $\Gamma = 0.1$ Hz and we showed that using Γ in the range of (0.01, 1) Hz does not significantly affect our results (see Note 5 in SI).

The system dependent vibrational frequencies, e.g., the Debye frequency, is a good measure to describe the attempt-frequency (Γ_0) (see Note 5 in SI). For that we used a Γ_0 value equal to the Debye frequency of SiC, i.e. 16 THz.

2. *I'm not sure I understand the argument made in the third paragraph on p. 9. The argument appears to be saying that the barriers decrease as the distance between monovacancies increases and that as a result nanoparticles should have smaller annealing temperatures than bulk due to the smaller associated distances. Shouldn't it be that if nanoparticles exhibit smaller distances between monovacancies than in the bulk, the annealing temperatures should be higher if the barrier decreases with increasing distance?*

We apologize for the lack of clarity. The barrier of V-V pairing to form VV decreases as the monovacancy (MV) separation decreases. However, we point out a special case where the barrier for the pairing of third (V-V³) neighbors is lower than that for the pairing of second neighbors (V-V²) vacancies. However, the barriers for the pairing of both V-V² and V-V³ to form VV are significantly lower than the barrier for distant V-V pairs, i.e. the V_{Si} migration barrier, where interactions between two MVs are weak.

We revised the manuscript on page 9:

“Our results indicate that in general, as V_{Si} and V_C approach each other, the barrier of V_{Si} migration towards V_C would decrease, facilitating the pairing of V-V to form VV.”

3. *The work appears to propose a general framework for investigating the synthesis of point-defects at the atomistic level, but the benchmarking focuses on a single defect the VV. Would it be possible to comment on whether or show that the formalism is indeed general (ie. by including more examples in 3C-SiC or other hosts, for example a comparison of the predicted annealing temperature for the formation of carbon vacancy-carbon antisite complexes in 3C-SiC and those found in the literature - <https://journals.aps.org/prb/abstract/10.1103/PhysRevB.101.184108>)*

We thank the reviewer for the suggestion. We added a discussion on the formation of the NV (N_CV_{Si}) center in SiC in the revised manuscript on page 12:

“In closing we note that the computational strategy adopted here to investigate the formation of VV and the results obtained in our work are useful to inform the investigation of other spin-defects in semiconductor hosts. For example, the formation of the N_CV_{Si} (denoted as NV) center in SiC is also expected to be facilitated by the migration of V_{Si} but in nitrogen doped samples [48, 49].

Therefore, we expect that the favorable E_F - T conditions identified here for the creation of VV should be partially applicable to the NV center as well. Indeed, an annealing $T \sim 1,200$ K was adopted experimentally for NV formation [48, 49], close to the predicted optimal value for the VV. Moreover, to guide and optimize the NV synthesis, our results can be used to estimate the desired E_F to stabilize V_C^0 instead of $V_C^{+1/+2}$, and hence mitigate the aggregation of NV (negatively charged for n-type conditions [50]) and V_C . In addition, our computational strategy can be used to compute the NV migration barrier to design the proper annealing temperature; our results suggest the need to elucidate the effects of nitrogen doping on the system E_F and on the induced reactions, e.g. the NV and Nc aggregation. Importantly, our results indicate that NV should be easier to form in hex-SiC than 3C-SiC, as the VV, due to the higher E_F .”

4. Further, regarding the generalizability of the results, would it be possible to comment on why the use of a PBE lattice constant is valid for obtaining barriers at the DDH level of theory. Essentially, the title appears a bit misleading if the first principles theory being used is not self-consistent.

Our first-principles molecular dynamics simulations in the NPT ensemble to predict the lattice expansion at finite temperatures were conducted at the PBE level of theory, as they would have been very demanding at the hybrid level of theory (DDH).

Based on the discussion of the lattice expansion effects in Note 5 in SI, the slight overestimate of the lattice constant by PBE would not affect the results reported in our paper.

In addition, we note the lattice constant of 4H-SiC at 0 K obtained with the PBE and DDH functionals differ by less than 0.5 %, as shown in TABLE S1 of *Phys. Rev. Materials* 5, 084603.

Therefore, we believe the use of the PBE lattice constant to obtain barriers at the DDH level of theory is justified.

5. Expanding on the previous comment, would it be possible to further elaborate on why we would expect error cancelations at the PBE and DDH levels of theory (from the supplementary material on p. 3 in the third paragraph)? (ie. Is it possible to show for some small supercell or for the unit cell that total energies calculated following perturbations to the atomic positions indeed exhibit the desired error calculations at the PBE and DDH levels of theory?)

The values of the entropy (S) of a defective system in the initial (IS) and transition (TS) states of a given path are determined by the respective phonon frequencies (ω). The entropy difference (∇S) between IS and TS is determined by the ratio of phonon frequencies (see Note 5 in SI). Assuming one mode for simplicity, we have $\nabla S = k_B \ln(\omega_{IS}/\omega_{TS})$, where k_B is the Boltzmann constant.

As shown in FIG. S5 of *Phys. Rev. Materials* 5, 084603, for pristine 4H-SiC, the phonon frequencies obtained with the PBE (ω^{PBE}) and DDH (ω^{DDH}) functionals are very close to each other: $\omega^{\text{DDH}} \approx 1.03 \omega^{\text{PBE}}$, and we expect a similar scaling for 3C-SiC.

When computing the phonon frequency ratio, the scaling factor (1.03) is canceled out. Hence, $\nabla S^{\text{DDH}} = k_{\text{B}} \ln(\omega_{\text{IS}}^{\text{DDH}} / \omega_{\text{TS}}^{\text{DDH}}) \approx k_{\text{B}} \ln(\omega_{\text{IS}}^{\text{PBE}} / \omega_{\text{TS}}^{\text{PBE}}) = \nabla S^{\text{PBE}}$

We added the following discussion to the revised SI on page 3-4:

“Let us consider one phonon mode for simplicity, and denote the phonon frequency (ω) at the initial (IS) and transition (TS) states as ω_{IS} and ω_{TS} , respectively. Then, ∇S can be computed as: $\nabla S = k_{\text{B}} \ln(\omega_{\text{IS}} / \omega_{\text{TS}})$ [1, 2]. Recent results [3] showed that the phonon frequencies obtained from PBE (ω^{PBE}) and DDH (ω^{DDH}) functionals for 4H-SiC are similar, with $\omega^{\text{DDH}} \approx 1.03 \omega^{\text{PBE}}$. It is reasonable to assume that such scaling relationship holds also in the case of 3C-SiC. Since the scaling factors entering the expression of ∇S cancel out, we can easily obtain ∇S at the DDH level of theory: $\nabla S^{\text{DDH}} = k_{\text{B}} \ln(\omega_{\text{IS}}^{\text{DDH}} / \omega_{\text{TS}}^{\text{DDH}}) \approx k_{\text{B}} \ln(\omega_{\text{IS}}^{\text{PBE}} / \omega_{\text{TS}}^{\text{PBE}})$, and we find $\nabla S^{\text{PBE}} \approx \nabla S^{\text{DDH}}$.”

Reviewer #3 (Remarks to the Author):

Silicon carbide (SiC) host prototypical color centers relevant for applications in quantum technology. The potential of SiC color centers not only hinges on the opportunities SiC provides as a mature electronic material, but also on the ability of their controlled intentional creation at the desired location to allow for single color center operation in place of ensembles.

The manuscript by Cunzhi Zhang Francois Gygi and Giulia Galli addresses the formation and annealing kinetics of prototypical color centers in Silicon carbide, namely the silicon vacancy, carbon antisite-vacancy complex, and the di-vacancy in SiC. They focus on the 3C polytype and investigate the activation barriers of prototypical mechanisms consisting of nearest neighbor and next neighbor hops in the frame work of density functional theory. The calculations involve the semilocal PBE-functional and the DDH hybrid functional and are conducted for using standard techniques for finding saddle points and energy barriers. For specific mechanisms they also include a finite temperature relevant for kinetic processes based on enhanced sampling within ab initio molecular dynamics.

We thank the reviewer for the comments. We emphasize that, as also recognized explicitly by the reviewer #1, the computational strategy presented here, especially the coupling of different methods, has never been applied before to study the formation of defects in semiconductors.

The formation and annealing kinetics of intrinsic defects in SiC has been investigated for more than twenty years both by theory and experiment and has seen renewed interest with SiC color centers becoming candidates for qubits and single-photon sources in particular as open questions

regarding vacancy complexes including the di-vacancy exist. The present manuscript follows an earlier work of the authors published last year in Nature Communications on the same topic focusing more on explorative ab initio molecular dynamics simulations.

We note that the earlier work was largely based on empirical force fields and did not present a detailed description of the various competing processes leading to the formation of double vacancies in SiC. In particular, our earlier work did not address the Fermi level and charge state dependence of defect processes at high temperatures (T), nor it discussed the importance of metastable states in the description of formation processes.

The present manuscript in large part confirms the results of earlier works regarding barriers and the Fermi level dependence of annealing mechanisms for the vacancies and the carbon antisite-vacancy complex adding additional insight in case of the di-vacancy. As key result they summarize this knowledge in a diagramm showing the activation temperature for specific formation or annealing mechanisms vs. the Fermi-level.

The reviewer is correct that some of our results are indeed consistent with previous studies. However, we emphasize that the results of earlier works regarding the barriers and Fermi level dependences of annealing mechanisms were largely *preliminary and qualitative*. The Fermi level engineering at high T , including effects of various defects, to achieve the optimal defect formation was never discussed before. The importance of metastable charge states in determining the effective barriers and defect dynamics was mostly ignored and not systematically considered in any study of the formation of defects in semiconductors.

Based on this diagramm and a discussion of the temperature dependence of the Fermi level as a function of defect density they conclude that an n-type Fermi level together with a certain temperature provide optimal conditions for the formation of quantum bits in 3C SiC.

Given that the present, well-written manuscript largely builds on the previous work of the authors already published in Nature communications and the earlier work by other groups, and the progress achieved I think the paper reflects the outcome of a regular article. In particular, important results of an in-depth of analysis of the formation and annealing kinetics concerning a prototypical qubit center, the NV-centers in diamond were published as a regular article. In this light I do not support the publication of the present work in Nature communications. Furthermore, the manuscript and conclusion contains a few issues that should be addressed:

As also recognized by reviewer #1, our work represents an important and novel step towards the systemic use of first-principles simulations to predict and design the optimal conditions for the synthesis of spin-defects in semiconductors, with many new results, relative to previous studies:

1. We propose a *computational protocol* integrating electronic structure calculations at 0 K and first-principles molecular dynamics at high- T , that is generally applicable to a variety of spin-defects and hosts. We demonstrate the effectiveness of the protocol by predicting optimal conditions for the formation of the VV in 3C-SiC, finding agreement with experiments, and revealing critical atomistic insights.
2. We quantify the defect behavior as a function of the Fermi level E_F and T and quantify the complex and interdependent role of multiple defects in tuning E_F and thus the defect dynamics. We believe such *quantitative predictions* (as opposed to partial qualitative results reported previously) are crucial for the *quantitative design* of spin-qubits and for the understanding and tuning of their synthesis conditions.
3. One important, fundamental result obtained in our study is the need to consider multiple charge states of defects and in particular thermodynamically meta-stable ones to obtain lower effective barriers for defect processes.

1) Although n-type conditions favor the formation or stability of the desired color centers, the conditions to stabilise the relevant active charge may not be compatible with n-type conditions. In the diagram in Figure 3 should also contain the information on the stability range of the relevant active charge states of the silicon vacancy and the di-vacancy. Only in the range of the Fermi level where the active charge state is stable, the proposed Fermi-level engineering provides an advantage. With this in mind

The formation of defects and the stabilization of desired charge-states are processes occurring at different conditions, specifically different temperature (T) conditions. For example, in SiC the processes leading to the formation of the VV occur at $T > 1,000$ K, while for most spin-defect applications the relevant T is 300 K or lower. Importantly, the E_F is strongly temperature dependent (see Fig. S4 and S5) and hence one cannot report the E_F range on the same diagram for the formation and charge-state stabilization processes. In our work, we discuss the engineering of the Fermi-level for the formation of defects at high- T , not the Fermi level to obtain the desired charge-state at low- T . The information about charged states of interest to the reviewer is directly available from Fig. S1 in SI.

However, the reviewer is correct that a discussion of the difference between the Fermi level required in formation and charge-state stabilization processes is in order and we added such a discussion in the revised manuscript on page 11:

“We note that the optimal E_F for the formation of defects (e.g. the VV) at high T may not coincide with the optimal value required to stabilize the desired charge state of the same defect at low T . Therefore, after formation, further engineering of E_F may be needed, e.g., to stabilize VV^0 or V_{Si}^- .”

2) In Fig. 3. the combination of nearby vacancies to di-vacancies with very low activation temperature is shown. Although these reactions certainly takes place at low activation temperature

it requires the generation of such pairs as a result of previous implantation - vacancy migration can be excluded as a source at these temperatures. Under these conditions nearby interstitials are also available and the recombination of vacancies with interstitials is a competing process. For this reason Fig 3. should also include the activation temperature for the interstitial migration and vacancy-interstitial recombination. The arguments in the discussion should be refined correspondingly

We agree with the reviewer that at low T , interstitials should be considered and may be relevant, e.g., to enable the recombination of vacancies and interstitials.

However, in our paper we do not aim at pointing out all possible processes occurring at low T , but rather that there also exist low- T processes leading to the formation of the VV (not only high- T ones) and we discuss, as an example, the pairing of V-V defects. A comprehensive investigation of competing, low- T processes involving interstitials is important, and can be a topic of a future study.

We added the following statement to the revised manuscript on page 10:

“However, it is worth mentioning that at lower T , the migration of interstitials that can then recombine with, e.g., V-V and VV may be relevant processes; additional studies are required for their elucidation.”

3) The authors argue that charge-state equilibrium occurs on timescales that are shorter than the timescale between successive migration events. The authors' estimate in support of this argument (cf. Note 3 of the supplementary material) does not include the dependence on the Fermi-level position of the carrier density, which is exponential. So while this argument is certainly true, when the free carrier density is high, vanishing free carrier densities in high purity semi-insulating SiC material does, indeed, lead to very slow recombination dynamics. In order to complement their argument the authors should plot the estimate for the recombination rate as a function of the Fermi-level position both for the 3C and 4H plotypes. In the expected case that there is a range of Fermi-levels

We agree with the reviewer that the charge-state equilibrium or the recombination dynamics of defects depend on the free carrier density and thus E_F . Following the suggestion of the reviewer, we estimated the carrier capture (emission) rate k (g) as a function of E_F (and position of defect level E_T within the gap) for 3C- and 4H-SiC. In the revised SI, we show the results in Fig. S3 and we added a discussion in Note 4.

Our results show that it is reasonable to assume a fast charge-state equilibration at high T , when computing effective barriers.

The addition to the SI (Note 4) is not reported here, due to the many equations and new figures mentioned above.

4) In Fig. 4 high concentrations of silicon vacancies are assumed as dominating defects at 1200K and 1400K. According to Fig. 3, however, the activation temperature for the conversion into the carbon antisite-vacancy complex is indicated to be below 1400K. This seemingly contradicts the conditions assumed in Fig 4. The main text does not provide in depth explanation in this respect. The authors should discuss the implications more rigourously in the text and resolve the seeming contradiction.

While directly after irradiation with high particle doses high vacancy concentrations are readily created, annealing at high temperature rapidly reduces these concentrations through interstitial-vacancy recombination or vacancy out-migration to thermal equilibrium. This can only be achieved by reducing the temperature below the activation temperature of vacancy migration. Is it reasonable to forcefully maintain a high vacancy concentration throughout the annealing to produce a higher di-vacancy concentrations and at the same time assume to obtain a high-grade material for quantum applications? The authors should answer this problem in the discussion of Fig. 4. I have doubts that the proposed route of Fermi-level engineering via high vacancy concentrations is a successful strategy.

The defect concentration in a given sample varies as a function of time during high T annealing. Vacancy concentrations can be high at some points during the annealing treatment, but then reduced or even close to zero as a function of time. However, quantifying the concentration of various defects as a function of annealing time is very challenging, and it depends on many parameters, including the initial defect distribution, concentration and types. Therefore, for simplicity, we considered several vacancies in a typical concentration range to obtain a qualitative estimate of their effects on the value of E_F .

We agree that the vacancy concentrations may be reduced or even increased, e.g., by V_{Si} to CAV conversion, after high- T annealing. In addition, maintaining high concentration of residual vacancies may degrade the material quality and thus its performances for quantum applications. Nonetheless, the results presented in Fig. 4 should be helpful to understand how to exploit the impact of specific defects on E_F , and to understand the defect dynamics during the high T annealing process, where vacancies and other complexes can be abundant and inevitable.

We added the following statement in the revised manuscript on page 11 as:

“We note that the concentration of vacancies varies with annealing time and vacancies may be easily annealed out at high T . In addition, large amounts of residual vacancies may degrade the material’s quality and hence its performance for quantum applications. Therefore, engineering E_F by means of a large vacancy concentration may not be a good strategy. Nonetheless, the results presented in Fig. 4 should be helpful to understand how to exploit the impact of specific defects

on E_F , and to understand the defect dynamics during the high T annealing process, where vacancies and other complexes can be abundant and inevitable.”

REVIEWER COMMENTS

Reviewer #1 (Remarks to the Author):

The authors have responded well to my comments and made the amendment I required. I suggest publication of the revised MS.

Reviewer #2 (Remarks to the Author):

The authors' insight that the consideration of metastable charge states of defects can lead to accurate first-principles predictions of optimal annealing temperatures makes the work exciting. Additionally, the authors have addressed all of my concerns regarding the paper. I would therefore support publication of the work in Nature Communications.

Reviewer #3 (Remarks to the Author):

In their revised manuscript the authors addressed the reviewers criticism and recommendation point by point. There is appreciable improvement of the manuscript, where the authors followed the reviewers suggestions. However, some crucial points have not been adequately dealt with and in the current version they may be even be scientifically misleading. The manuscript clearly needs further revision.

The open points concern the following:

1. Reviewer#3 addresses in his/her first point the notion of Fermi-level engineering in conjunction with the stability of the

active qubit states. In their revision, they seem to have understood the high relevance of this point for the validity of the proposed strategy as a main result of their manuscript. However in writing

"... We note that the optimal EF for the formation of defects

(e.g. the VV) at high T may not coincide with the optimal value

required to stabilize the desired charge state of the same defect at

low T. Therefore, after formation, further engineering of EF may be needed, e.g., to stabilize VV^0 or VSi^{\dots} "

they fall short of a true discussion and rather try to divert the

reader from a critical point. In order to maintain the proposed

general strategy they should honestly discuss available strategies for

what they call "... further engineering of EF...". The relevance of the whole strategy hinges on whether such engineering is realistic or not.

Device structures with gates may be an option. Further doping or

processing steps on the other hand would require additional annealing steps that eventually also will affect (degrade) the defect abundance including that of the desired qubit centers.

The case of NV^- centers in SiC, suggested by reviewer#2, is

exceptional as the defect itself favours n-type doping with nitrogen for its formation. Although it supports the authors thesis, it actually is not transferable to the investigated case of intrinsic centers, namely the di-vacancy.

2. Reviewer#3 addresses in the second point the notion of

interstitial-vacancy recombination and requests a thorough discussion of this point in the context of the proposed recombination of neighbouring vacancies.

By noting "... However, it is worth mentioning that at lower T, the migration of interstitials that can then recombine with, e.g., V-V and VV may be relevant processes; additional studies are required for their elucidation." they vaguely indicate the possibility of such processes without making a relevant

link to a vacancy-clustering that is activated in a temperature range where the vacancy-interstitial recombination is an adverse effect and again diverting this relevant point. There is sufficient discussion of the vacancy-interstitial recombination in the literature e.g. in Ref 44. or the work by Gao,

Weber et al. such that a more elaborated discussion could be provided including activation temperatures taken from the literature.

3. Reviewer#3 in the fourth point request a resolution of the

contradiction between Fig. 3 and Fig. 4. regarding the suggestion that high silicon vacancy concentrations are suggested to stabilize the Fermi-level at temperatures above their annealing temperatures (activation of the conversion or migration that would lead to rapid out diffusion of vacancies in excess of the natural abundance). In the present version of the manuscript still high excess concentrations of e.g. silicon vacancies are present and seemingly suggested as a means to engineer the Fermi-level. I request that the authors should remove all possible defect-scenarios from the corresponding panels of Fig.4, where the temperature exceeds the activation temperature of the defect for the specific Fermi level position indicated in Fig. 3. This includes V_{Si} , V_c , and CAV .

4. Reviewer#3 request a revision of the point on thermal equilibrium

between individual migration events and the notion of the effective

barrier. The authors have vastly improved the discussion of this

point in the supplementary note 4. However, there are two

issues. First in the comparison of the timescales of carrier

capture/recombination with the one of migration events for the latter

only one case with an 3eV barrier is used yielding a time between

diffusion events of 10s. If one tries a barrier heights of 2eV or

smaller, this yields a timescale close to $\sim 10^{-4}$ s or faster. This

carrier recombination timescale for intrinsic conditions is just a

factor 10 longer than times shown for 3C-SiC and close to

4H-SiC. Taking the temperature dependence of σ with an activation

energy for multi-phonon processes into account, the recombination

timescale easily may increase by an order of magnitude or

two. Therefore at least in 4H-SiC the timescales for charge state equilibration and kinetic processes may interfere and the former is not necessarily warranted. I request that the authors should work on this point when they want to make any prediction/discussion of 4H-SiC. I suggest to present a version of Fig. 3 than in the critical region for 4H does not make use of the effective barriers.

Response to reviewers

We thank the Reviewer for the thoughtful suggestions and comments. We respond below (in blue text) to all the additional comments of Reviewer # 3 and we point out (in red text) all the revisions applied to the manuscript.

All changes relative to the previous manuscript and SI are shown in blue.

Reviewer #3 (Remarks to the Author):

In their revised manuscript the authors addressed the reviewers criticism and recommendation point by point. There is appreciable improvement of the manuscript, where the authors followed the reviewers suggestions. However, some crucial points have not been adequately dealt with and in the current version they may be even be scientifically misleading. The manuscript clearly needs further revision.

The open points concern the following:

1. Reviewer#3 addresses in his/her first point the notion of Fermi-level engineering in conjunction with the stability of the active qubit states. In their revision, they seem to have understood the high relevance of this point for the validity of the proposed strategy as a main result of their manuscript. However in writing

"... We note that the optimal EF for the formation of defects (e.g. the VV) at high T may not coincide with the optimal value required to stabilize the desired charge state of the same defect at low T. Therefore, after formation, further engineering of EF may be needed, e.g., to stabilize VV⁰ or VSi⁻..."

they fall short of a true discussion and rather try to divert the reader from a critical point. In order to maintain the proposed general strategy they should honestly discuss available strategies for what they call "... further engineering of EF...". The relevance of the whole strategy hinges on whether such engineering is realistic or not.

Device structures with gates may be an option. Further doping or processing steps on the other hand would require additional annealing steps that eventually also will affect (degrade) the defect abundance including that of the desired qubit centers.

The case of NV⁻ centers in SiC, suggested by reviewer#2, is exceptional as the defect itself favours n-type doping with nitrogen for its formation. Although it supports the authors thesis, it actually is not transferable to the investigated case of intrinsic centers, namely the di-vacancy.

Our original statement “[...] *Therefore, after formation, further engineering of E_F may be needed* [...]” is a statement referring to the general strategy that one may want to adopt for different point defects and host materials. We emphasize that for the case studied here (VV in 3C-SiC) the optimal conditions of VV formation at high temperature (T) are compatible and consistent with the conditions required to stabilize VV^0 at low T . Hence our proposed “Fermi-level (E_F) engineering (see Fig. 4 and text in the manuscript)” is justified.

Specifically, we note that Fig. 8 of [*PHYSICAL REVIEW B* 96, 085204 (2017)], Fig. 2 of [*PHYSICAL REVIEW B* 92, 045208 (2015)] and our own results shown in Fig. S1 (obtained using the high- T lattice constant of 3C-SiC), show that VV^0 is stable at n -type conditions in 3C-SiC. In our paper, we propose to use **n -type conditions** via, e.g., N_C or V_C to facilitate VV formation at high T (see Fig. 4 and text in the manuscript). At low T , the presence of N_C is still responsible for n -type conditions (Fig. S4A), in the region where VV^0 is stable; in addition, the residual number of V_C present after annealing could pin the E_F to ~ 2 eV (near the V_C (+1/0) charge-transition-level; Fig. S4B), i.e. at values for which again VV^0 is stable.

However, as shown in Fig. 8 [*PHYSICAL REVIEW B* 96, 085204 (2017)] and Fig. 2 [*PHYSICAL REVIEW B* 92, 045208 (2015)], in 4H-SiC VV^0 is only stable at near intrinsic conditions. These results are again consistent with our findings that near **intrinsic conditions** are favorable for the formation of VV at high T in 4H-SiC (see text in the manuscript).

It is important to note the difference in conditions required for *both* the formation and the charge-state stabilization of the defect in the different polytypes of SiC; these conditions are different in 3C- and 4H-SiC; specifically, they are n -type and intrinsic conditions, for 3C- and 4H-SiC, respectively. This difference emphasizes that the specific strategy to engineer the E_F depends on the specific qubit and host. Importantly, the computational scheme presented in our work can be generally used to design such “specific strategy for E_F engineering for a qubit” for a variety of systems and spin-defects, as demonstrated here for the specific example of VV in SiC.

With these observations in mind, we come again to comment on our original statement “[...] *Therefore, after formation, further engineering of E_F may be needed* [...]”. It is conceivable that there may be hosts and/or spin-defects, where the required doping conditions for the optimal qubit formation at high T and its charge-state stabilization at low T are different. This is the reason why we wrote “... *after formation, further engineering of E_F may be needed* ...”. However, once more, it is important to recognize that our computational scheme can be used to design optimal conditions for qubit formation at high T . As suggested by the reviewer, “gating or additional doping” maybe used to “further engineer the E_F ” to stabilize the qubit to the desired charge-state at low T .

We added a discussion of these points to the revised manuscript on page 11:

“We observe that the proposed n -type conditions for efficient VV formation at high T , is also favorable to stabilize the desired VV^0 at low T ; the presence of, e.g., N_C or V_C could increase the E_F (see Fig. S4) to a region where the VV^0 is stable [48, 49]. However, we note that it is conceivable that there may be spin-defects or hosts, where the required doping conditions for the optimal qubit formation at high T and its charge-state stabilization at low T are different. In those cases, after qubit formation, further engineering of E_F may be needed via, e.g., gating or additional doping.”

and on page 12:

“We note that, to synthesize VV in hex-SiC, it is beneficial to use samples at near intrinsic conditions, where the VV^0 is stable [48, 49].”

2. Reviewer#3 addresses in the second point the notion of interstitial-vacancy recombination and requests a thorough discussion of this point in the context of the proposed recombination of neighbouring vacancies.

By noting “... However, it is worth mentioning that at lower T , the migration of interstitials that can then recombine with, e.g., V-V and VV may be relevant processes; additional studies are required for their elucidation.” they vaguely indicate the possibility of such processes without making a relevant link to a vacancy-clustering that is activated in a temperature range where the vacancy-interstitial recombination is an adverse effect and again diverting this relevant point. There is sufficient discussion of the vacancy-interstitial recombination in the literature e.g. in Ref 44. or the work by Gao, Weber et al. such that a more elaborated discussion could be provided including activation temperatures taken from the literature.

We would like to point out that the results in Ref. 44 [*PHYSICAL REVIEW B* 69, 235202 (2004)] are obtained using the LDA functional, and that in [*Journal of Applied Physics* 94, 4348 (2003)] the authors used empirical potentials. Although we know that when using advanced functionals such as hybrid functionals (as DDH in our paper), results for barriers may differ from LDA results, and although we know that empirical potentials may not be as accurate as DFT calculations, we revised our discussion of interstitial-vacancy recombination on page 10, by using the results of the two papers quoted by the reviewer:

“However, it is worth mentioning that the recombination of close interstitial-vacancy complexes and the interstitial migration process could involve low barriers, $< \sim 1.5$ eV in SiC, according to Ref. [44, 47]; using these barriers the activation temperatures for these processes are estimated to be $< \sim 500$ K, i.e. within a range of T where the pairing of V-V to form VV may also occur. Hence, we expect the recombination of V-V with interstitials and its pairing to be competing processes below 500 K, and interstitials may have the adverse effect of reducing the VV formation from V-V pairing.”

3. Reviewer#3 in the fourth point request a resolution of the contradiction between Fig. 3 and Fig. 4. regarding the suggestion that high silicon vacancy concentrations are suggested to stabilize the Fermi-level at temperatures above their annealing temperatures (activation of the conversion or migration that would lead to rapid out diffusion of vacancies in excess of the natural abundance). In the present version of the manuscript still high excess concentrations of e.g. silicon vacancies are present and seemingly suggested as a means to engineer the Fermi-level. I request that the authors should remove all possible defect-scenarios from the corresponding panels of Fig.4, where the temperature exceeds the activation temperature of the defect for the specific Fermi level position indicated in Fig. 3. This includes V_{Si} , V_C , and CAV.

We agree with the reviewer that over long timescales (i.e. over times reaching the end of the annealing cycle), using defects to engineer the E_F at a temperature higher than the activation temperature for conversion or migration, is not a viable strategy.

However, within shorter timescales (during the defect formation at high T), certain defects may be present at a temperature higher than their respective activation temperature, e.g., that for migration and sometimes it is inevitable to find such defects in the sample. Under these circumstances, although gradually annealed out, the presence of defects would affect the position of E_F . To characterize such “transient” effects on E_F , in Fig. 4 we include the results for relevant vacancies, i.e. V_{Si} , V_C and CAV.

For instance, when considering annealing T of 1300K, the V_{Si} migration and the V_{Si} to CAV conversion are activated processes. At the beginning of the annealing, V_{Si} may still be present in the sample. Moreover, V_{Si} migration, e.g., to the sample surface, V_{Si} conversion to CAV and V_{Si} migration to aggregate with V_C and form VV should all occur. In other words, the whole VV formation process will be affected by the presence of V_{Si} , V_C and CAV. As a result, quantifying the effects of V_{Si} , V_C and CAV on E_F is necessary and justified.

To clarify this point, we revised our discussion on page 11:

“Nonetheless, at the beginning of the annealing treatment, vacancies can be present even at temperatures higher than their respective activation temperature for migration or conversion. Under these circumstances, the presence of vacancies would definitely affect the position of E_F . To characterize and further understand the impact of such transient species during the annealing process, we include the results for relevant vacancies at high temperatures in Fig. 4.”

4. Reviewer#3 request a revision of the point on thermal equilibrium between individual migration events and the notion of the effective barrier. The authors have vastly improved the discussion of

this point in the supplementary note 4. However, there are two issues. First in the comparison of the timescales of carrier capture/recombination with the one of migration events for the latter only one case with an 3eV barrier is used yielding a time between diffusion events of 10s. If one tries a barrier heights of 2eV or smaller, this yields a timescale close to $\sim 10^{-4}$ s or faster. This carrier recombination timescale for intrinsic conditions is just a factor 10 longer than times shown for 3C-SiC and close to 4H-SiC. Taking the temperature dependence of σ with an activation energy for multi-phonon processes into account, the recombination timescale easily may increase by an order of magnitude or two. Therefore at least in 4H-SiC the timescales for charge-state equilibration and kinetic processes may interfere and the former is not necessarily warranted. I request that the authors should work on this point when they want to make any prediction/discussion of 4H-SiC. I suggest to present a version of Fig. 3 than in the critical region for 4H does not make use of the effective barriers.

Following the suggestion, we added one figure to the SI (Fig. S6), which shows results obtained without using effective barriers.

We added discussion on page 12:

“We note that the charge-state equilibration process is slower in hex- than 3C-SiC (see Fig. S3). Therefore, in Fig. S6, we show defect transformation barriers determined from the most stable charge-state at a given E_F . By comparing Fig. 3 and Fig. S6, we find that our qualitative predictions of the VV formation properties in hex-SiC are the same, whether using effective barriers or the data of Fig. S6.”